# ECOSTRESS Reveals the Importance of Topography and Forest Structure for Evapotranspiration from a Tropical Forest Region of the Andes

Alejandra Valdés-Uribe [1,*], Dirk Hölscher [1,2] and Alexander Röll [1]

1 Tropical Silviculture and Forest Ecology, University of Goettingen, Büsgenweg 1, 37077 Göttingen, Germany; dhoelsc@gwdg.de (D.H.); aroell@gwdg.de (A.R.)
2 Centre of Biodiversity and Sustainable Land Use, University of Goettingen, Platz der Göttinger Sieben 5, 37073 Göttingen, Germany
* Correspondence: dvaldes@uni-goettingen.de

**Abstract:** Tropical forests are major sources of global terrestrial evapotranspiration (ET), but these heterogeneous landscapes pose a challenge for continuous estimates of ET, so few studies are conducted, and observation gaps persist. New spaceborne products such as ECOsystem Spaceborne Thermal Radiometer Experiment on Space Station (ECOSTRESS) are promising tools for closing such observation gaps in understudied tropical areas. Using ECOSTRESS ET data across a large, protected tropical forest region (2250 km$^2$) situated on the western slope of the Andes, we predicted ET for different days. ET was modeled using a random forest approach, following best practice workflows for spatial predictions. We used a set of topographic, meteorological, and forest structure variables from open-source products such as GEDI, PROBA-V, and ERA5, thereby avoiding any variables included in the ECOSTRESS L3 algorithm. The models indicated a high level of accuracy in the spatially explicit prediction of ET across different locations, with an r$^2$ of 0.61 to 0.74. Across all models, no single predictor was dominant, and five variables explained 60% of the models' results, thus highlighting the complex relationships among predictor variables and their influence on ET spatial predictions in tropical mountain forests. The leaf area index, a forest structure variable, was among the three variables with the highest individual contributions to the prediction of ET on all days studied, along with the topographic variables of elevation and aspect. We conclude that ET can be predicted well with a random forest approach, which could potentially contribute to closing the observation gaps in tropical regions, and that a combination of topography and forest structure variables plays a key role in predicting ET in a forest on the western slope of the Andes.

**Keywords:** remote sensing; elevation; aspect; leaf area index; machine learning; mountain forest; Ecuador; GEDI; PROBA-V; ERA5

## 1. Introduction

Forests dissipate heat through evapotranspiration (ET), which is a crucial biophysical process in sustaining the hydrological cycle and regulating climate [1,2]. As such, ET impacts rainfall patterns by "recycling" precipitation [1,3,4] and attenuates temperature extremes at local and regional levels [1,5]. ET is influenced by complex interactions between environmental variables, species compositions, and forest structure [1,6]. Over the last decades, an intensification of the global hydrological cycle with increases in global ET has been observed and attributed with high confidence to human activities related to changes in land use [7,8]. Tropical forests are main sources of global terrestrial ET [9], and their continuing transformation into other types of land use affects ET, with potentially severe consequences for regional and global climates which may further intensify global warming [1,2,10]. Although accurate ET estimations across tropical forests are pivotal in improving our understanding of future climate responses [11,12], only a few ET studies

are available for tropical regions thus far due to their remoteness and often challenging accessibility [9,13].

New spaceborne experiments such as the ECOsystem Spaceborne Thermal Radiometer Experiment on Space Station (ECOSTRESS) can help cope with such challenges in tropical regions. ECOSTRESS is a thermal radiometer coupled to the International Space Station that measures land surface temperature between ~52°N and ~52°S. The ECOSTRESS algorithm derives different levels of data products, one of which is the Level-3 ET estimate calculated from the PT-JPL algorithm [14]. It is available as an instant ET value at the respective time of overflight and as an extrapolated daily ET estimate. Compared to established remote sensing products such as Landsat or MODIS, ECOSTRESS provides data with higher spatial (70 m × 70 m) and temporal resolutions (1–5 days) [15]. The ECOSTRESS Level-3 ET product has been validated with measurements from 82 eddy covariance sites, showing a close linear relationship across different biomes and land use types ($r^2$ = 0.89, [15]). Although tropical ecosystems were strongly underrepresented due to a lack of eddy sites, the good performance of ECOSTRESS across different regions is attributed to the efficiency of the PT-JPL model applied to calculate ET, paired with the high accuracy of the land surface temperature measurements [16]. While further studies are needed to better assess and validate the performance of ECOSTRESS across tropical (forest) ecosystems, existing studies show that ET can be estimated well by the PT-JPL algorithm across tropical biomes in South America ($r^2$ > 0.6) [17] and in a tropical mountain forest at the eastern Andes [18], suggesting that the ECOSTRESS L3 ET product does closely reflect actual spatial ET patterns and is thus a powerful asset for studying the magnitude, spatial variability, temporal dynamics and driving mechanisms of ET along diverse environmental gradients. Such studies are further facilitated by an increasing array of openly available global satellite products characterizing local topographies [19], climates [20], and ecosystem structures [21–23], often with similarly high spatial resolutions as ECOSTRESS.

The influences of physical and biological factors on the spatial variability of ET and across (tropical) forest ecosystems are diverse and locally dependent [1,24,25]. Therein, day-to-day variations in ET at a given site are often explained by fluctuations in microclimatic variables such as air humidity, wind speed, and solar irradiance [26,27], while seasonal ET variations in the (sub)tropics are often related to water availability [28]. Previous studies have related spatial variability in ET to topographic variables such as elevation, slope, or aspect [29,30], as well as to variables related to ecosystem structure [31]. In contrast to topographic characteristics, ecosystem structure variables are highly variable across time, often because of human intervention. For example, tropical forests typically have higher canopy cover with larger trees, and higher leaf area indexes than agroforestry or agricultural systems [32,33]. Changes in ecosystem structure associated with forest conversion or degradation thus often result in substantial changes in key vegetation–atmosphere interactions such as ET [34–36].

While modern remote sensing experiments provide products with higher spatial and temporal resolutions, observation gaps remain. Particularly in tropical (mountain) regions where partial or complete cloud cover is frequent, the availability of data from remote sensing platforms is limited [37]. Machine learning approaches are potentially powerful tools for filling existing observation gaps, e.g., in ECOSTRESS-derived ET maps. Machine learning algorithms can be trained to predict ET or other ecosystem exchange processes from freely available ancillary variables (e.g., [20–22]) over thousands of pixels and can subsequently be applied to close observation gaps via spatially explicit predictions. To date, several different machine learning algorithms exist, and they are widely applied throughout the environmental sciences to analyze large datasets that potentially involve dozens of interacting variables, complex non-linear relationships, and several thousand observations [38–40]. RF is one of the most popular algorithms and often performs well in ecological contexts, e.g., when applied to predict reference ET, water stress, sap flux, leaf stomatal conductance, net ecosystem exchange, or changes in land cover [41–46]. The application of machine learning for spatial predictions is a special case, and in recent years,

some specific studies have addressed autocorrelation issues to avoid overfitting [47–49], which is mainly caused by a lack of independence between training and testing (prediction) samples [50]. Techniques such as forward feature selection and target-oriented cross-validation with a random forest (RF) approach reduce the risk of spatial overfitting and demonstrate more realistic model performances [47,48,51].

In our study, we used an advanced RF machine learning approach to predict ET for 2250 km$^2$ of tropical forest (>80% tree cover, [21]) located on the western slopes of the Andes, inside Cotacachi Cayapas National Park in Ecuador. We combined daily ET estimates from ECOSTRESS with freely available topographic, meteorological, and forest structural remote sensing products. We aimed to determine (1) what combination of variables is relevant for the spatial prediction of ET across the region, and (2) what roles forest structure variables play in the spatial prediction of ET.

## 2. Methodology

### 2.1. Study Area

This study was carried out in Cotacachi Cayapas National Park, a protected area located in northwestern Ecuador (Figure 1). The park hosts 62% of the total native plant species in Ecuador [52] at the intersection of two global biodiversity hotspots, the Tropical Andes and the Chocó-Darién Ecoregion [53]. It extends over 2610 km$^2$ [54], and its elevation ranges from 34 to 4989 m a.m.s.l. [19]. The mean annual air temperature ranges from 2 °C to 25 °C, and the annual precipitation ranges from 1076 mm to 3251 mm [55,56]. In the lowlands, the dry season begins in June and extends until September, while at higher elevations, it can last until November [52]. During the wettest month, precipitation ranges between 143 mm and 408 mm, and during the driest month, precipitation ranges between 21 mm and 165 mm [56]. Along the altitudinal gradient, local reports describe eight different ecosystems, the most dominant of which are evergreen premontane forests (49% of the total area), evergreen low montane forests (20%), evergreen montane forests (10%), and lowland rainforests (3%) [54]. We focused our study on 2250 km$^2$ (86%) of the national park that corresponded to a forested area with more than 80% tree cover [21] and elevations of up to 4055 m [19].

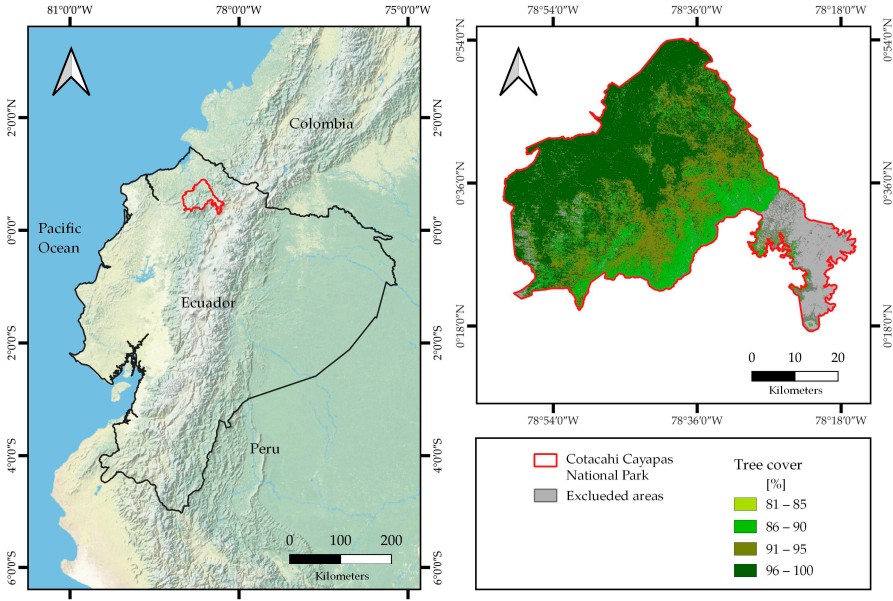

**Figure 1.** Location of Cotacachi Cayapas National Park in northwestern Ecuador (**left**). Park boundaries, excluded areas (<80% tree cover according to [21], e.g., tussock grass), and tree cover across 2250 km$^2$ of forest in the study area (**right**).

### 2.2. Study Design and Data Collection

The target variable in our study was the daily evapotranspiration (ET) from the ECOsystem Spaceborne Thermal Radiometer Experiment on Space Station (ECOSTRESS), which is available as a Level-3 (L3) latent heat flux estimate in W m$^{-2}$ with a spatial resolution of 70 m × 70 m [15]. The daily ET estimate is derived from the instant ET value (at the time of overflight) by adjusting the data to the diurnal radiation intensity cycle and assuming that the relative ratios of the instantaneously measured variables are constant throughout the daytime [57]. The data were retrieved for the entire year 2020 through the ECOSTRESS Early Adopter Program and transformed into geo-located grid data using the Python *swath2grid* programming tool [58]. Daily ET values were transformed from W m$^{-2}$ into mm d$^{-1}$ for better comparability to other studies [11], therein assuming 12 h of daylight for our study region, which is in close proximity to the equator. For the year 2020, 15 images were available for the study region, but only a few met the selection criteria of (1) images captured during the daytime, i.e., between 9:00 and 17:00 UTC-5, (2) ET estimates associated with less than 70% uncertainty (based on the uncertainty layer "ETinst Uncertainty" provided with the ECOSTRESS L3 product), and (3) the number of pixels covering the area, i.e., an image covering at least 50% of the study area. Three images taken on 6 February, 8 August, and 10 October met all three criteria, covering between 58 and 66% (1290–1446 km$^2$) of the study area (note that none of the images covered the entire area of interest). On day 1 (D1), 6 February 2020, the image comprised 301,072 pixels (corresponding to a forest area of 1446 km$^2$); on D2 (8 August 2020), it comprised 268,085 pixels (1290 km$^2$); and on D3 (10 October 2020), the image comprised 271,694 pixels (1307 km$^2$).

As the basis for the prediction of ET across the study area and for each of the three days, a set of physical and biological variables was retrieved from freely available remote sensing products. We excluded all variables that were directly part of the calculations of the ECOSTRESS L3 algorithm (net radiation, soil heat flux, air temperature, vapor pressure deficit, relative humidity, surface reflectance in the red band, surface reflectance in the near-infrared band, normalized difference vegetation index, and soil adjusted vegetation index), and related variables were retrieved from different independent satellite products [19–22], leading to a final dataset of five topographic, four meteorological, and four forest structure variables (Table 1).

Using QGIS 3.16 [59], data were projected to the same coordinate system (EPSG:32717–WGS 84/UTM zone 17S) and aligned to the same extent. Due to large differences in spatial resolution among the predictor variables, the ECOSTRESS data were transformed into points, and the *point sampling tool* plugin was used to extract the information from the respective predictor raster layers for each ECOSTRESS pixel-point location. Topographical data were derived from the Global Digital Surface Model ALOS World 3D product (version 3.2) generated by the Japanese Aerospace Exploration Agency (JAXA), which has a 30 m resolution [19]. QGIS was used to calculate the slope, aspect, and topographic position index (TPI). As aspect is a circular variable, it was transformed into two distinct continuous variables (ranging from 0 to 1): eastness and northness [60]. Hourly meteorological data between 6:00 and 18:00 UTC+5 was retrieved from the European Centre for Medium-Range Weather Forecast (ECMWF) Reanalysis v5 (ERA5), which has a spatial resolution of 9 km [20]. The extracted variables were the 10 m u-component of wind, 10 m v-component of wind, dew point temperature, skin reservoir content, and total precipitation. The variable wind speed was derived from the u and v components, as shown in Equation (1). All retrieved ERA5 data were aggregated to daytime means using the raster calculator tool in QGIS.

$$ws = sqrt\,(u^2 + v^2) \tag{1}$$

**Table 1.** Target variable, evapotranspiration, and predictor variables in the categories of topography, meteorology, and forest structure, as extracted from different open-access remote sensing products.

| Variable Category | Variable Description | Variable Abbreviation | Unit | Remote Sensing Product | Original Resolution |
|---|---|---|---|---|---|
| Target variable | Observed daily evapotranspiration | ET | mm d$^{-1}$ | ECOSTRESS [15] | 70 m |
| Topographic | Derived data from digital surface model (DSM) | Elevation<br>Slope<br>Eastness<br>Northness<br>TPI | m<br>$^\circ$<br>rad<br>rad<br>m | ALOS World 3D-JAXA [19] | 30 m |
| Meteorological | Wind speed calculated from u and v components<br>Dew point temperature<br>Skin reservoir content<br>Total precipitation | Ws<br><br>DewT<br>SkRc<br>Prec | m s$^{-1}$<br><br>$^\circ$K<br>mm<br>mm | ERA5-ECMWF [20] | 9 km |
| Forest structure | Tree cover | TreeCov | % | Global forest change 2000–2020—Landsat [21] | 30 m |
| | Tree height | TreeH | m | Global forest canopy height—GEDI and Landsat [22] | 30 m |
| | Leaf area index | LAI | m$^2$ m$^{-2}$ | PROBA-V V1 [23] | 300 m |
| | Roughness length | SfRo | m | ERA5 and ECMWF [20] | 28 km |

Forest height data were retrieved from a global forest canopy height map at a spatial resolution of 30 m (the data were recorded between 18 April 2019 and 2 October 2019) [22]. Tree canopy cover data were retrieved from the global forest cover map at a 30 m spatial resolution, using the most updated version, version 1.9 (the forest cover baseline calculated in the year 2000 and the forest change from the year 2000 to 2021) [21]. Roughness length was extracted from the previously mentioned ERA5 product at a spatial resolution of 28 km [20]. Leaf area index (LAI) data for 2020 were extracted from the Project for On-Board Autonomy—Vegetation (PROBA-V) V1 at a spatial resolution of 300 m (the data were recorded between 15 July 2019 and 31 March 2020) [23]. As the study area is covered by evergreen forest types, the LAI was assumed to be constant throughout the year of study. The three datasets analyzed in our study (one for each of the three studied days) comprised the daily ET estimates from ECOSTRESS and the respective data on the 13 predictors for each pixel (Table 1). A preliminary quantitative analysis showed significant linear relationships between most of the predictors and ET, but only marginal parts of the variance in ET were explained ($r^2$: < 0.001–0.055, $p < 0.05$) (Figures S1–S3).

*2.3. Statistical Analysis*

To decrease the risk of overfitting and to obtain models with the ability to accurately predict across different locations, we applied the best-practice strategies outlined in previous studies [47,61,62]. Such strategies are time- and computing-intensive processes, and we thus limited our study to a single machine learning algorithm. The random forest (RF) approach was selected for the spatial prediction of ET, as observed from ECOSTRESS with 13 potential predictors (Table 1). The RF approach combines multiple decision trees from a randomly selected sample of the training data and different randomized combinations of explanatory variables to predict a target variable. Each decision tree produces independent new values corresponding to every subgroup of the training data. The final prediction is a single value that averages all predictions [63]. A particular strength of RF is that it can determine the contributions of single variables in explaining the studied phenomena [64,65], making it a popular algorithm in ecological studies. To account for spatial autocorrelation and to avoid overfitting, we applied three main strategies: (1) the

spatial block approach, as suggested in [62], to assign a spatial ID, (2) RF forward feature selection (FFS) with *k-fold* leave-location-out cross-validation (LLO-CV) to select the best spatial predictors [51,66], and (3) RF modeling with *k-fold* LLO-CV [47]. Spatial blocks were created using the R package *blockCV* [67], FFS was carried out with the *CAST* package [47], and RF was implemented in the *randomForest* package [68] via the *caret* package [69].

We first divided the study area into spatial blocks of 2 km$^2$ (Figure 2), i.e., each pixel within a given block was identified with a unique spatial ID between 1 and 5. The data were randomly split into training (40%) and testing (60%) sets before the FFS. The FFS algorithm works by combining all pairs of variables. It saves the best initial model and adds further variables, with a function capturing model improvement as each variable is added using LLO-CV [47]. FFS is a time-intensive process; thus, as suggested in [61], a sub-sample of 50,000 pixels from the training dataset was selected using stratified random sampling and the spatial ID as splitting criterion. Based on the spatial ID and using the *CreateSpacetimeFolds* function from the *CAST* package, the subsample was divided into five folds *(k-fold = 5)* for spatial cross-validation [47]. The combination of predictor variables selected at each split (*mtry*) was set at a value of two. For the 13 predictors in our study, the FFS algorithm thus evaluated a maximum of 576 models. The best variables selected by the FFS were used as input predictors for the final RF model for each of the studied days (D1–D3). Following the workflow applied in [61], all training data—which were previously assigned to a spatial fold based on the spatial ID—were used for tuning and final training. The number of predictor variables selected at each split (*mtry)* was tuned for each model between two and the maximum selected number of predictors after FFS. The number of trees were tuned between 200 and 2500 (200 to 1000 in increments of 100 and 1000 to 2500 in increments of 500) [65]. The tuning process showed that for all three models, the best *mtry* value was two, and the best number of trees was 1000.

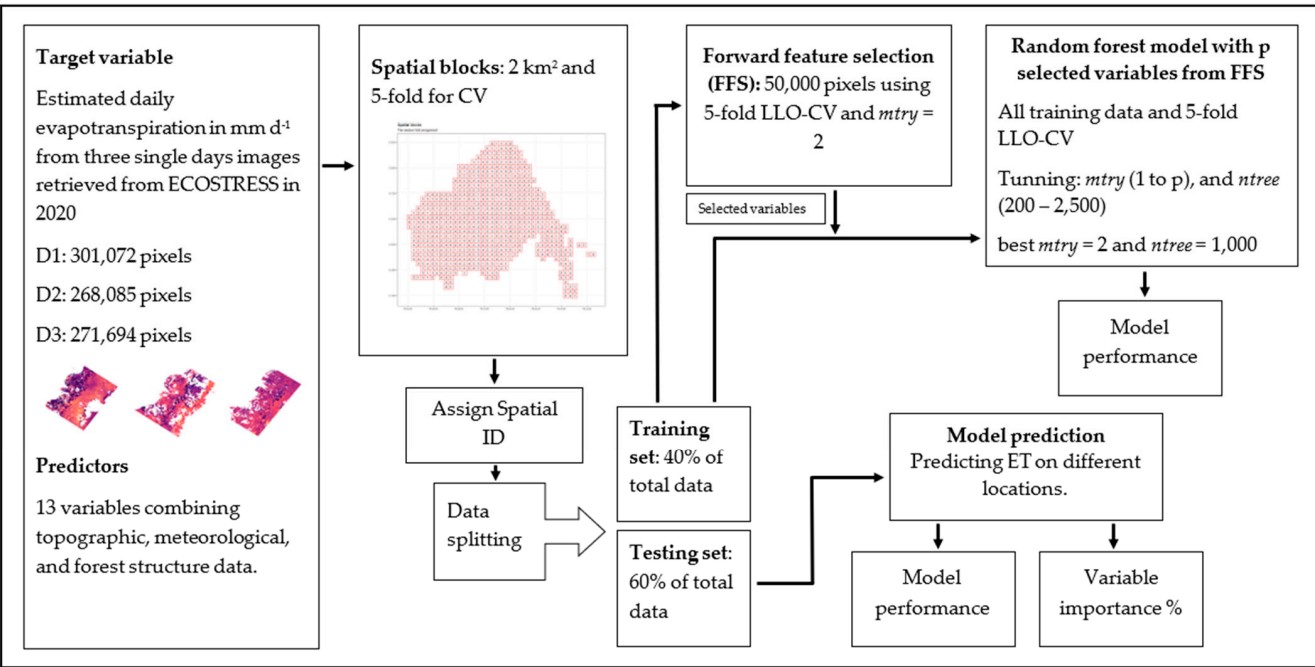

**Figure 2.** Data splitting, variable selection, and modeling applied to the datasets comprising ECOSTRESS data and topographic, meteorological, and forest structure variables. Methodology adapted from [61].

Model performances were analyzed by predicting over the independent testing dataset, which comprised 60% of the total data. We calculated the mean absolute error (MAE) and root mean squared error (RMSE) in mm d$^{-1}$ as well as the normalized root mean square error (nRMSE) in % for the model training and testing steps. We further calculated the

coefficient of determination, $r^2$, as an indicator of the variance explained by each model. To assess the contribution of each predictor to the final model outcome, the individual variable importance (in %) was extracted using the wrapper *varImp* function from the R package *caret* [69]. All statistical analysis and plotting were carried out with R 4.2.2 [70], and the maps were produced in QGIS [59].

## 3. Results

The forward feature selection (FFS) of topographic, meteorological, and forest structure variables for the prediction of ET resulted in different combinations of variables for each of the studied days (D1–D3) (Table 2). For D1, 10 of the 13 available variables were selected, while 11 variables were selected for D2 and D3. The FFS removed slope, precipitation (Prec), and roughness length (SfRo) from D1; for D2, it removed northness and skin reservoir content (SkRc); and for D3, it removed dew point temperature (DewT) and tree canopy cover (TreeCov). The remaining variables were used to model ET via a random forest (RF) approach with spatial cross-validation (CV). There were only marginal differences in the mean square error (MAE), root mean square error (RSME), normalized root mean square error (nRMSE), and $r^2$ values between the respective training and testing outcomes (Table 3), indicating that the models can predict ET for locations that were not part of the model training. Overall, model performance was good, with $r^2$ values of 0.61 to 0.74, MAE values between 0.3 mm d$^{-1}$ and 0.6 mm d$^{-1}$, and an nRMSE below 14%.

**Table 2.** Topographic, meteorological, and forest structure variables after forward feature selection (FFS) as the basis for modeling ET with a RF approach on the studied days (D1–D3). TPI: topographic position index; Ws: wind speed; DewT: dew point temperature; SkRc: skin reservoir content; Prec: total precipitation; TreeCov: tree cover; TreeH: tree height; LAI: leaf area index; SfRo: roughness length.

| Model/Date | Variable Set from FFS | | |
|---|---|---|---|
| | **Topographic** | **Meteorological** | **Forest Structure** |
| **D1: 6 February 2020** | Elevation, eastness, northness, and TPI | Ws, DewT, and SkRc | TreeCov, TreeH, and LAI |
| **D2: 8 August 2020** | Elevation, eastness, slope, and TPI | Ws, DewT, and Prec | TreeCov, TreeH, LAI, and SfRo |
| **D3: 10 October 2020** | Elevation, eastness, northness, slope, and TPI | Ws, SkRc, and Prec | TreeH, LAI, and SfRo |

**Table 3.** Comparison of model accuracies in terms of mean absolute error (MAE), root mean square error (RMSE), normalized root mean square error (nRMSE), and coefficient of determination ($r^2$) for training and testing (prediction) datasets for the studied days (D1–D3).

| | **Training** | | | | **Testing** | | | |
|---|---|---|---|---|---|---|---|---|
| | **MAE** | **RMSE** | **nRMSE** | **$r^2$** | **MAE** | **RMSE** | **nRMSE** | **$r^2$** |
| | mm d$^{-1}$ | mm d$^{-1}$ | % | - | mm d$^{-1}$ | mm d$^{-1}$ | % | - |
| **D1: 6 February 2020** | 0.48 | 0.67 | 11.54 | 0.73 | 0.49 | 0.67 | 11.51 | 0.74 |
| **D2: 8 August 2020** | 0.56 | 0.75 | 13.38 | 0.62 | 0.54 | 0.76 | 13.33 | 0.62 |
| **D3: 10 October 2020** | 0.32 | 0.49 | 8.33 | 0.62 | 0.32 | 0.49 | 8.32 | 0.61 |

The average observed ET values for over 160,000 ECOSTRESS pixels for each of the studied days were 5.8 ± 1.3 mm d$^{-1}$ (D1), 5.7 ± 1.2 mm d$^{-1}$ (D2), and 5.9 ± 0.8 mm d$^{-1}$ (D3) (mean ± SD, Figure 3). The predicted ET means were nearly identical to the observed means, with divergences of less than 0.1%. The predicted minima were consistently higher than the observed minima, and the predicted maxima were lower than the observed maxima, resulting in ranges that were 25% to 33% smaller and standard deviations that were 23% to 33% smaller for the predicted ET compared to the observed ET. The predicted data were also characterized by a less multimodal distribution than the observed data (Figure 3).

The spatial patterns of ET in the study area predicted from the RF models resembled the observed ET patterns well on all days (Figure 4). There were no apparent patterns in the distribution of divergences (differences between observed ET values and predicted ET values for the testing datasets) across the study area. Pixel-level divergences between observed and predicted ET values ranged from $-4.11$ mm d$^{-1}$ to 4.18 mm d$^{-1}$ for D1, from $-4.06$ mm d$^{-1}$ to 3.32 mm d$^{-1}$ for D2, and from $-3.75$ mm d$^{-1}$ to 3.63 mm d$^{-1}$ for D3 (Figure 4). Between 84% and 95% of the predicted ET values had divergences less than or equal to 1 mm d$^{-1}$ (Figure 4). Scatter plots of ET divergences vs. all predictors revealed no clear influences of the studied environmental variables on the accuracy of the spatial ET predictions. Interestingly, while there were no trends in the average divergences when plotted along different environmental gradients, the maximum observed divergences partially show distinct non-linear patterns, e.g., with maximum ET divergences at intermediate values of elevation, precipitation, or wind speed (Figures S4–S6).

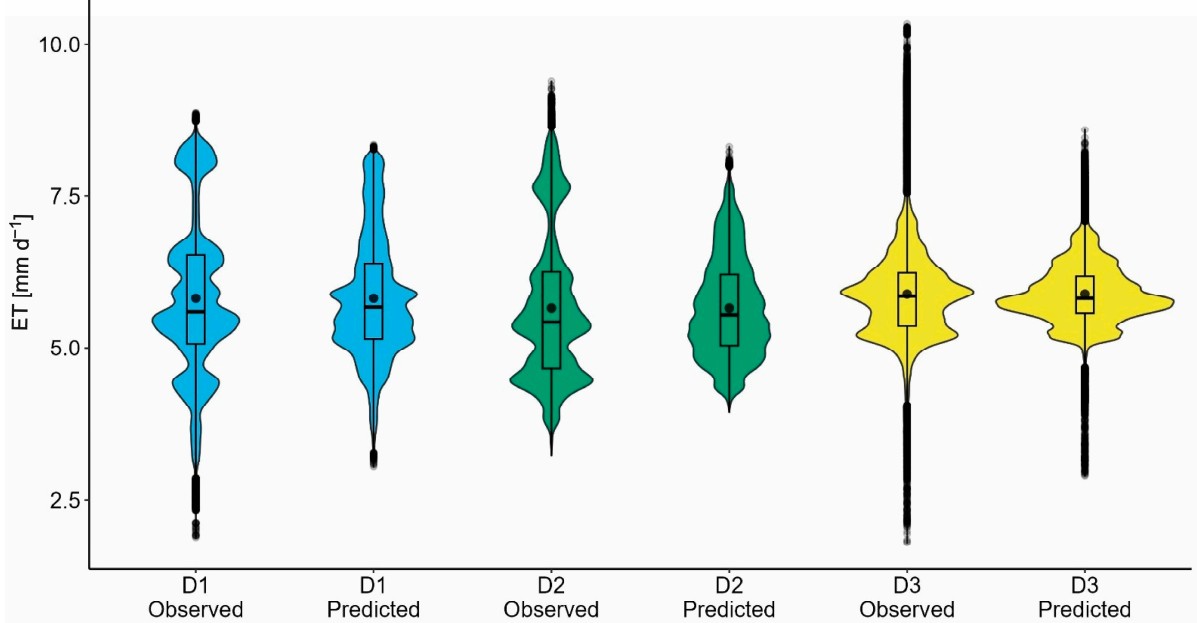

**Figure 3.** Violin plots showing the observed vs. predicted daily ET (mm d$^{-1}$) for the testing datasets: D1: 6 February 2020 (predictions over 180,643 pixels), D2 on 8 August 2020 (predictions over 160,851 pixels), and D3 on 10 October 2020 (predictions over 163,016 pixels). Observed ET was extracted from ECOSTRESS L3 daily ET [15]. Predicted ET was derived from an RF algorithm using a set of topographic, meteorological, and forest structure variables.

An analysis of variable importance for the RF modeling on the studied days showed that it takes five variables to explain at least 60% of the results (Figure 5). The three most important variables in all models were elevation, eastness (topography), and LAI (forest structure), in varying order and with varying contributions to the model outcome (12–14% for elevation, 10–16% for eastness, and 13–14% for LAI). Further variables with contributions of more than 10% in at least one model were northness and tree height (D1), slope (D2), and roughness length (D3), i.e., more variables of topography and forest structure. Overall, the respective sets of topographic variables together explained between 43% and 47% of the spatial predictions of ET, the forest structure variables explained between 30% and 31%, and the meteorological variables explained between 19% and 25%.

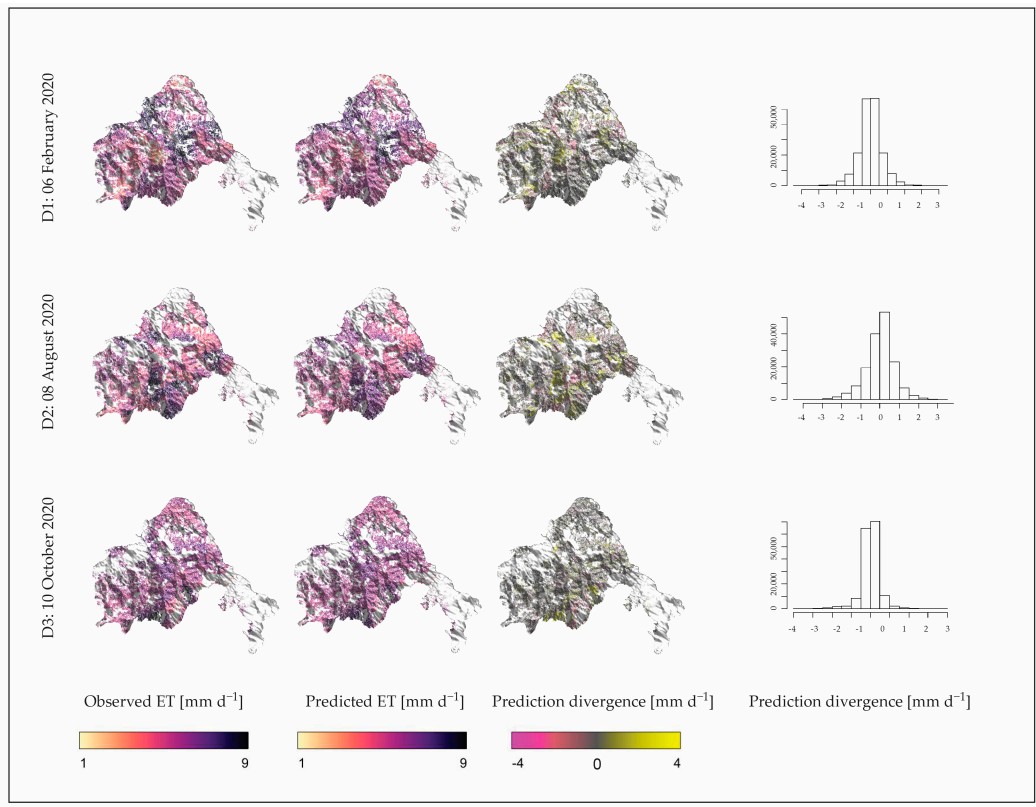

**Figure 4.** Spatial distribution of observed ET, predicted ET, and prediction divergences in Cotacachi Cayapas National Park for the studied days (D1–D3). D1: 6 February 2020 (predictions over 180,643 pixels), D2 on 8 August 2020 (predictions over 160,851 pixels), and D3 on 10 October 2020 (predictions over 163,016 pixels). White areas on the maps correspond to excluded pixels (forest coverage < 80%, missing data from the predictor variables), training data or no data (observation gaps from ECOSTRESS). Frequency distribution of prediction divergencies (right column) show that 87% of pixels for model D1, 84% for model D2, and 95% for model D3 had divergence values less than or equal to 1 mm d$^{-1}$.

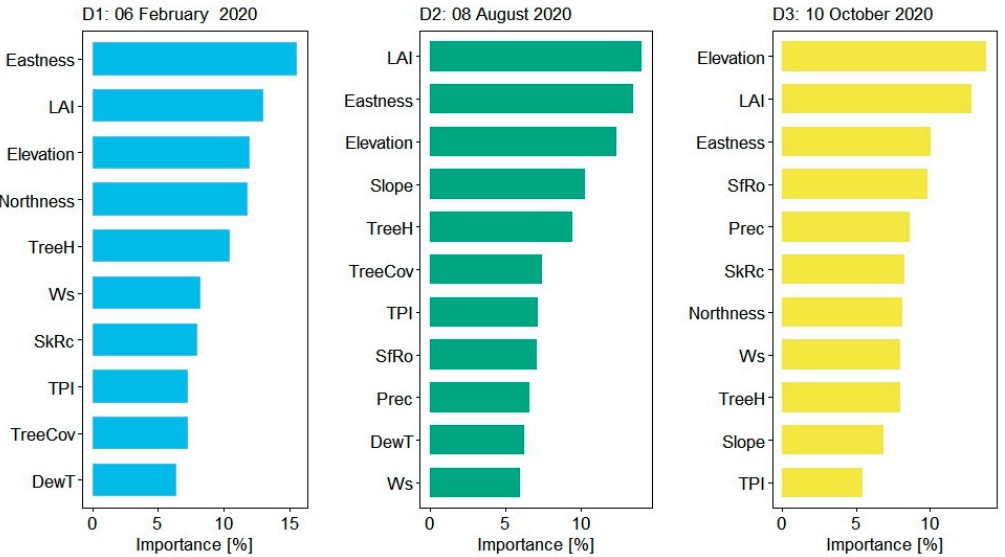

**Figure 5.** Variable importance (% of contribution to model outcome) for spatial ET predictions using a RF approach for the studied days (D1–D3). TPI: topographic position index; Ws: wind speed; DewT: dew point temperature; SkRc: skin reservoir content; Prec: total precipitation; TreeCov: tree cover; TreeH: tree height; LAI: leaf area index; SfRo: roughness length.

## 4. Discussion

### 4.1. Observed ET Data from ECOSTRESS

There are no previous ET assessments for the study region for comparison, but our daily means fall into the range of values reported for tropical evergreen forests in Indonesia and Costa Rica (on average, 4.1 mm d$^{-1}$–6.3 mm d$^{-1}$) [71,72]. They are also comparable to a recent study in a tropical mountain forest in southern Ecuador in which ET was assessed with a scintillometer and a daily ET value of up to 6 mm d$^{-1}$ was reported [73]. The same study reported large day-to-day variations in ET (1 mm d$^{-1}$–6 mm d$^{-1}$), which were attributed to differences in solar irradiance between sunny, overcast, and rainy days. In our study, only three days in 2020 met all quality criteria, including a large, cloud-free coverage of the study area. This may explain the similarly high values across the days in our study, which were distributed relatively evenly over the year (February, August, and October 2020). The equatorial study region generally has high precipitation, no substantial dry periods, and is dominated by evergreen forest types; the lack of seasonal variation in the observed daily ET across the relatively cloud-free study days thus seems reasonable. In our study, we selected only the ECOSTRESS ET estimates with less than 70% uncertainty to balance data quality and data availability, as an exploratory analysis showed severely limited data availability when applying stricter uncertainty thresholds. This demonstrates that even with state-of-the-art remote sensing methods, obtaining high-quality, low-uncertainty estimates of ET in tropical (mountain) regions remains a challenge. In the Appalachian Mountains, a study along an elevation gradient reported a close linear relationship between ECOSTRESS ET and eddy covariance (r$^2$ = 0.64, *p* < 0.05, [74]). The study also reported a weaker but significant relationship when including all estimates regardless of the associated ET uncertainty (r$^2$ = 0.27, *p* < 0.05). In the study in the Appalachian Mountains, the ECOSTRESS ET data were consistently higher than the ground-based reference values [74], which may partially explain the high observed maximum daily ET values in our study area.

### 4.2. Predicting ET from Random Forest Modeling

Forward feature selection (FFS) was applied to reduce the risk of spatial overfitting and to improve spatial model performance [47]. It reduced the dataset for ET prediction from the original 13 variables (five topographic, four meteorological, and four forest structure variables) to a final set of 10 to 11 variables. Even though the FFS method generally tends to remove geolocation variables, the final sets of predictors comprised such variables (e.g., elevation and topographic position index). Likely, our training data were well distributed across the large study area with complex and heterogeneous topography so that the spatial clustering that leads to variable exclusion in FSS did not occur [47]. Due to the spatial nature of our data, we applied a target-oriented cross-validation strategy (LLO-CV) [48,61,62]. The very similar values of MAE, RSME, nRMSE, and r$^2$ between training and testing (prediction) outcomes suggest that spatial overfitting did not occur in the models and that they can thus predict ET for locations that were not part of the model training. The ability of our models to predict ET at different locations was further applied exemplarily in filling observation gaps (see maps in Figure S7). Overall, the performance of the applied random forest (RF) algorithm was good across the studied days, with RMSE values ranging from 0.49 mm d$^{-1}$ to 0.67 mm d$^{-1}$, nRMSE values ranging from 8% to 13%, and model prediction accuracy values (r$^2$) of 0.64 to 0.74. These accuracy metrics are comparable to other studies applying RF modeling for the spatially explicit prediction of ecological target variables, such as reference ET (nRMSE = 9%, r$^2$ = 0.91) [42], daily ET (RMSE = 1.27 mm d$^{-1}$, r$^2$ = 0.63) [75], sap flux (r$^2$ = 0.8), and stomatal conductance (r$^2$ = 0.5) [43].

In our study, the RF models yielded precise mean ET values compared to the observations, but the predictions of extreme values were softened by the algorithm, with 25–33% smaller ranges in predicted ET than in observed ET. Likewise, the distributions of the predicted datasets were less multimodal than the observed data, with most predicted ET values occurring between 3 mm d$^{-1}$ and 7 mm d$^{-1}$. While it is well known that machine learning algorithms tend to favor the most typical data, from an ecological point of view,

less frequent observations with extreme values may carry important information [76]. There is room for improvement in future machine learning applications to better account for the importance of rare observations in ecological studies, e.g., by assigning a degree of importance to observations and by applying enhanced methods to distinguish between outliers and minority data and by using ensemble learning algorithms [76,77]. Overall, our study helps demonstrate that RF is a powerful tool for the spatial prediction of key ecosystem exchange processes such as ET, especially in tropical areas where satellite-based observation gaps persist. Our study is a good prospective for future assessments over extended areas at the western Andes to generate continuous ET coverage maps based on the available ECOSTRESS L3 ET data.

### 4.3. Variable Importance

The analysis of variable importance in the RF models showed that no single variable was dominant (<15% contribution to model outcome), i.e., the ET spatial prediction is driven by multiple variables and their interactions. Together, topographic variables contributed more to model outcomes (43% to 47%) than meteorological (19% to 25%) and forest structure variables (30% to 31%). The three most important single variables in all models were elevation and eastness (topography) and leaf area index (LAI, forest structure).

The strong effect of the topographic variables on ET spatial predictions are within expectations and in line with a recent study in southeastern Ecuador in which spatial ET variability was explained by elevation and topographic position [73]. In our study, the variables elevation (12–14%) and eastness (aspect) (10–16%) had major roles across all models. Along large elevation gradients, such as those in our study (34 to 4055 m a.m.s.l.), major shifts in forest structure and composition occur in response to changing environmental conditions [78,79]. Likewise, eastness is known to affect forest distribution and composition in the Ecuadorian Andes [80] and thus potentially influences ET. For example, individuals of the same species growing on shaded slopes can show different responses to water stress than species located on sunny slopes [81], leading to distinct ET patterns and dynamics [82,83]. Slope had intermediate variable importance (7–10%) in two out of the three models and was removed by the FFS in the third model. Intermediate slopes can accommodate more trees per unit ground area than level areas, while past a certain threshold, slopes become too steep to support forest ecosystems [31,84], thus potentially influencing ET via changes in vegetation cover and tree density. In complex mountainous terrain, slope may additionally affect wind flow regimes and thus energy and water exchange [85]. It also affects soil water availability, which can constrain or enhance local ET; as such, previous studies reported enhanced (evapo)transpiration on slopes compared to waterlogged valley bottoms [77,83]. The variable topographic position index was present in all three models but was of relatively low variable importance (5–7%). It is likely mainly of relevance in the ET prediction of pixels with extreme positions, such as on ridges or in depressions, with largely contrasting topographic characteristics to neighboring pixels. Some of the studied topographic variables may (partially) act as proxies for the environmental conditions at a given pixel, i.e., solar radiation, air temperature, or other climatic factors that are known to influence ET. In our study, as key drivers of ET, some climatic variables, particularly radiation, were excluded from the analysis for statistical reasons, i.e., because they were directly used in the ECOSTRESS L3 algorithm. This may partially explain the high importance of topography as a potential radiation proxy variable in our study. In general, the use of topographic variables in ET prediction models, e.g., for the purpose of filling the spatial gaps on ET maps with partial cloud cover, has advantages over the direct use of climatic variables and the use of most forest structure variables. As such, topographic variables are relatively simple to calculate and extract, are available at a much higher spatial resolution, are consistent over large timespans and, perhaps most importantly, are available with no data gaps. The latter may be a substantial advantage when, e.g., attempting to fill ET observation gaps in regions with frequent cloud cover, such as in mountainous (tropical) regions.

Among the four meteorological variables that were not part of the ECOSTRESS L3 algorithm and were thus used in RF modeling, wind speed occurred in all three models, with relatively low contributions (6–8%) to model outcome. In general, wind speed is an important driver of hydrological processes and is a key variable in understanding ET dynamics [86,87]. A higher wind speed often enhances ET (up to critical thresholds) by accelerating the turbulent exchange of water vapor from the leaves to the atmosphere in the boundary layer [87,88]. However, previous studies reported that aerodynamic properties in tropical regions influence less than 20% of ET and between 20% and 35% of ET in the sub-tropics [87,89]. In mountainous regions, wind systems are often of particular importance; for example, on the steep slopes of the western Andes, wind uplift causes orographic precipitation on the windward side and rain shadow effects on the lee side where foehn winds create dry valleys [90,91]. The lack of a strong effect of wind speed in our study may (partially) be related to the relatively coarse spatial resolution of the ERA5 satellite product. Two further meteorological variables which each occurred in two out of the three models with relatively low contributions (7–9%) were related to water availability, i.e., precipitation and skin reservoir content. Several previous studies reported that water availability can severely restrict ET when passing certain thresholds, both under conditions of water surplus [31,92,93] and water scarcity [94–96]. The non-dominant role of variables relating to water availability in our study, in addition to the consistently high mean ET values observed for the studied days, likely indicate that water availability was not a strongly limiting factor. As a further meteorological variable, dew point temperature occurred in two out of the three models with low contributions to model outcome (6%). It is related to air pressure, humidity, and precipitation and thus indirectly influences ET. A potential reason for its low contribution is that precipitation and skin reservoir content, i.e., variables more directly expressing ecosystem water status, already adequately integrate ET responses to varying water availability.

In our study, we focused on the roles of forest structure variables for spatially explicit predictions of ET. The variable LAI occurred in all three models and was the most important (14% for D2) and second most important variable (13% for D1 and D3). Leaves mediate the exchange of gas between ecosystems and the atmosphere, and LAI thus plays a key role in the carbon and water cycle and often is a key input variable in ecological modeling [97,98]. LAI varies substantially over space and time, and a higher LAI is associated with higher ET [36]. In line with our results, previous studies reported the LAI to be a key predictor of the spatial variability of transpiration in tropical ecosystems of the Americas [99] and ET in a mountain cloud forest [31]. We speculate that the importance of LAI for predicting ET in Cotacachi Cayapas National Park may be even higher than reflected in our results; the applied LAI product—the best product available for the study area in terms of coverage and spatial resolution—still had a spatial resolution 10 times lower (300 m, [23]) than, e.g., the topographical set of variables and thus does not adequately reflect smaller-scale LAI heterogeneity in the field [45,100]. Future LAI and thermal infrared products with finer resolutions to better characterize the biophysical properties of vegetation could further improve the accuracy of ET predictions over broad forest areas and across different land use types and support the generation of total ET coverage maps, which could help researchers study response patterns along larger environmental gradients. Further studied forest structure variables were tree height (intermediate contributions to all three models, 8–10%), surface roughness length (intermediate contributions to two models, 7–10%) and tree cover (relatively low contributions to two models, 7%). Increases in vegetation height and cover are often associated with higher ecosystem biomass and greater leaf area index and thus higher levels of ET [101,102]. Vegetation height and surface roughness length, the latter of which characterizes the heterogeneity of vegetation height, also influence ET via their roles in the turbulent energy and water exchange by shaping the ecosystem–atmosphere boundary layer [103].

The greater contribution of topographic variables to the model outcomes in our study compared to the contribution of forest structure variables is, in part, related to

the experimental design. In our study, we provided baseline results from near-undisturbed forests. Thus, we only considered forests with a tree canopy cover of more than 80% (as assessed from the global tree cover map in [21] at a spatial resolution of 30 m), excluding degraded forests and the tussock grass vegetation in the national park. This strongly reduces the spatial variability in the studied forest structure variables as intact forests are expected to generally have much higher LAI, canopy cover, and tree height values than neighboring degraded forests or agricultural systems [32,33]. Additionally, the temporal variability in forest structure and particularly LAI in our study region was also rather low due to the lack of strong seasonality and the associated dominance of evergreen forests, as well as the lack of higher-spatio-temporal-resolution data available for the area of study. Forest structure variables would likely provide a more dominant contribution to spatial predictions of ET across (tropical) seasonal forests or across heterogeneous landscapes with different land use types, which makes an interesting subject for follow-up studies in other previously underreported tropical areas. We consider our study in the evergreen forests of the western Ecuadorian Andes as a first attempt at a spatially explicit prediction of ET based on a set of freely available topographic, climatic, and forest structure variables and as a potential solution to fill observation gaps. There is much room for further improving spatial predictions of ET (or other ecological variables) with a new generation of remote sensing products with new and enhanced sensors and platforms, steadily increasing spatial and temporal resolutions and increasingly automated and sophisticated data extraction, quality control, and gap-filling procedures [104–106]. Potential improvements from a software/algorithm perspective could involve semi-automated and guided data handling and extraction approaches (at best, from multiple products simultaneously), as well as a standardized set of recipes (with pre-set quality filters e.g., for automated cloud removal and uncertainty thresholds) and associated gap-filling procedures.

## 5. Conclusions

We used a random forest approach to predict the daily evapotranspiration (ET) across a tropical forest region on the western slopes of the Ecuadorian Andes. Using a set of remotely sensed predictors, we achieved fair errors and good model performances. The spatial predictions of ET were mostly influenced by topographic and forest structure variables. Therein, the elevation, aspect, and leaf area index (LAI) were consistently the most important set of variables. We hypothesize that finer-spatio-temporal-resolution LAI products can further improve ET predictions and will contribute to generate wall to wall maps across the western Andes, where multiple types of land use converge.

**Supplementary Materials:** The following supporting information can be downloaded at: https://www.mdpi.com/article/10.3390/rs15122985/s1, Figure S1: Relationship between ET as retrieved from ECOSTRESS L3 ET product and all 13 predictor variables for D1; Figure S2: Relationship between ET as retrieved from ECOSTRESS L3 ET product and all 13 predictor variables for D2; Figure S3: Relationship between ET as retrieved from ECOSTRESS L3 ET product and all 13 predictor variables for D3; Figure S4: Relationship between divergencies and predictor variables for D1; Figure S5: Relationship between divergencies and predictor variables for D2; Figure S6: Relationship between divergencies and predictor variables for D3; Figure S7: Prediction of ET based on models D1: 06 February 2020, D2: 08 August 2020 and D3: 10 October 2020 for observation gap filling in Cotacachi Cayapas National Park.

**Author Contributions:** Study conceptualization: A.V-U., A.R. and D.H.; data collection: A.V-U.; data analysis and interpretation: A.V.-U., A.R. and D.H.; writing—first draft: A.V-U.; writing and editing: A.R. and D.H. All authors have read and agreed to the published version of the manuscript.

**Funding:** Thanks to the German Academic Exchange Service (DAAD) for funding my doctoral studies. We acknowledge support from the Open Access Publication Funds of Göttingen University.

**Data Availability Statement:** The data presented in this study are available upon request from the corresponding author.

**Acknowledgments:** We thank all reviewers for their valuable and accurate contributions.

**Conflicts of Interest:** The authors declare no conflict of interest.

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
