# Peer review of "ECOSTRESS Reveals the Importance of Topography and Forest Structure for Evapotranspiration from a Tropical Forest Region of the Andes"

_remotesensing, doi:10.3390/rs15122985_

Round 1

Reviewer 1 Report (Previous Reviewer 1)

Line 55 to 67: To much technical detail is given here for an introduction. I would suggest to shorten this paragraph a bit without mentioning the statistics from Reference 15.

Line 105 to 107: The sentence about time consumption is neither necessary here, nor does it play a major role for the present study. I would suggest to remove it.

Line 112f: What do you mean by "influence". Please specify.

Figure 1: There is a "2.2 Study design and data collection" the end of the caption (l. 134). Maybe only a pet peeve from my side, but I would suggest making the legend box as wide as the box above and not use Degree Minute Seconds format for the coordinates but Decimal Degree.

Line 160ff: You could put more emphasis on the fact that the variables you actually use for the modelling are independent of cloud cover and hence valuable for gap filling.

Line 219: Consistent wording please. Is "feature" here a "variable"?

Line 220: "Hence" is not necessary and misleading here. Please remove.

Line 226: the randomForest package is not "part of the caret package". Please rephrase.

Figure 2: This Figure would really benefit from some maps, pictures, etc. something to guide the eye. Also the font is different, Boxes are not aligned, arrows are different types and sizes. This can be done much better to improve readabilty. As it is an important figure for the reader I think some work is worth the effort.

Line 55 to 67: To much technical detail is given here for an introduction. I would suggest to shorten this paragraph a bit without mentioning the statistics from Reference 15.

Line 105 to 107: The sentence about time consumption is neither necessary here, nor does it play a major role for the present study. I would suggest to remove it.

Line 112f: What do you mean by "influence". Please specify.

Figure 1: There is a "2.2 Study design and data collection" the end of the caption (l. 134). Maybe only a pet peeve from my side, but I would suggest making the legend box as wide as the box above and not use Degree Minute Seconds format for the coordinates but Decimal Degree.

Line 160ff: You could put more emphasis on the fact that the variables you actually use for the modelling are independent of cloud cover and hence valuable for gap filling.

Line 219: Consistent wording please. Is "feature" here a "variable"?

Line 220: "Hence" is not necessary and misleading here. Please remove.

Line 226: the randomForest package is not "part of the caret package". Please rephrase.

Figure 2: This Figure would really benefit from some maps, pictures, etc. something to guide the eye. Also the font is different, Boxes are not aligned, arrows are different types and sizes. This can be done much better to improve readabilty. As it is an important figure for the reader I think some work is worth the effort.

Author Response

Reviewer 2 Report (Previous Reviewer 2)

The authors did a good job addressing my comments. I’m happy to recommend acceptance of the paper.

Note for L486, it would be both LAI and thermal infrared (TIR) / land surface temperature (LST) for ET, in addition to meteorology. I also liked your comment to me on the software/algorithm perspective—I think it would be useful for readers for you to include that in the paper as well.

I still think Fig 4 would be more powerful if draped on an underlying topo basemap, but this absence does not preclude publication of the paper.

L199. “prelaminar”  “preliminary”

Author Response

Reviewer 3 Report (Previous Reviewer 3)

The authors have satisfactorily addressed the issues I pointed out in the first review.

Author Response

Dear reviewer, thank you for taking the time to assess our replies and revise the manuscript. We are pleased to know that we addressed have addressed your concerns. Thank you again for your comments, which have helped us to further improve our work.

This manuscript is a resubmission of an earlier submission. The following is a list of the peer review reports and author responses from that submission.

Round 1

Reviewer 1 Report

The presented study in the manuscript "ECOSTRESS reveals the importance of topography and forest structure for evapotranspiration from a tropical forest region of the Andes" models evapotranspiration derived from the ECOSTRESS sensor with topographic, meteorological and structural predictors with the machine learning algorithm random forest. Through the internal variable importance of the random forest model, the study tries to find the contributions of the predictors to evapotranspiration in the study area. The random forest model is tuned and validated with contemporary methods from the field of spatial modelling such as spatial cross validation and variable selection.

While the premise of the study is appealing and new - the novel ECOSTRESS sensor combined with different products to analyse drivers of tropical ecosystems, I have several issues with the study design and logic of the manuscript:

1. The chosen modelling strategy is not appropriate for the goals of the manuscript. Random Forest (or a machine learning model in general) and the chosen model validation strategies are designed for (spatial) predictions. Although a spatial prediction is shown in Figure 4, it is not clear to me how and why this prediction was made. For the stated goal of identifying drivers of evapotranspiration, a spatial prediction is not necessary.

   Recommendation: If such a prediction is shown,
   a) introduce it: Why is such a prediction useful? I.e. why not simply use the ECOSTRESS product in the first place? Are there usability constrains of the sensor?
   b) discuss it: Are there interesting spatial patterns of evapotranspiration? Are these patterns the same in the observed and predicted evapotranspiration? Are there reasons why the model is better or worse in a particular region?

2. In the conclusion it is stated that "Spatial ET variability was mostly explained by topographic and forest structure variables" (l. 468f). However the shown variable importance is based on models using all training data (l. 211). This means that variable importance is based on the model fit on the training data (regardless of spatial CV or previous spatial feature selections) and does not reflect an explanations of spatial variability. While this could be an appropriate strategy to answer the raised question of explaining evapotranspiration, the model can simply be based on all available data (without training/test split). As a consequence, this weakens the decision to use spatial block CV and spatial evaluation.

3. In the methods section, several important points are missing or are too vague. How was the data assigned to one of the 5 folds (l. 200)? How was the data split into 40 / 60 train and test sets (l.201)? I assume not every single number between 200 and 2500 trees was tuned (l.213). Why is R² "an indicator of spatial variability explained by the model?" (l.219). The caret varImp function  does not calculate the contributions of the predictors (as stated in l. 220ff). Variable importance is calculated by the randomForest function, carets varImp is just a wrapper around the importance function of the randomForest package, which in turn just calls the variable importance that was calculated during training (see help pages of caret::varImp and randomForest::importance)

4. The first part of the discussion (l. 299 - l. 336) feels very unrelated to the study. Highlight your main findings, mention the novelties. At this point in the manuscript, arguing that the response variable from ECOSTRESS is valid  (on which the whole study relies) is too late. Information provided here better fits in different parts of the manuscript. L. 299 - 301 are results, L. 315 - 335 is better suited in the introduction to give reason why a use of ECOSTRESS might be promising.

Despite these flaws, I still think the stated idea of evapotranspiration analysis with ECOSTRESS and associated environmental variables is very promising. If the scope and aims of the study are redefined and shifted (either towards spatial predictions of evapotranspiration or more towards explaining evapotranspiration patterns with environmental variables) the study has a good chance to be published in the future. I hope this review helps in refining the study and gave some additional thoughts to consider.

Reviewer 3 Report

The reading of this manuscript was a rather interesting experience. As far as I can see, the study was technically conducted in a way that was practically flawless. On the other hand, the aims and the logical structure of the manuscript left me quite perplexed. I see the logic of the manuscript as follows.

The authors use a set of remotely sensed features to explain, through a random forest (RF) model, variation in evapotranspiration that is estimated for a certain day via an algorithm that uses a different set of remotely sensed features. There are three days for which estimates of evapotranspiration (ET) are available. The resulting three RF models explain 61–74% of the variation in ET and consistently the most important variables (in descending order) were topographical, forest structural and meteorological.

The numerical analyses used to get the results are quite clear and performed technically correctly. However, I really don't know what one can learn about or do with these results. In the Conclusions, it's stated that the study improves knowledge on ET and its environmental controls in western Andean forests. There is no direct expression of what this improved knowledge more precisely is but, special attention is given to leaf area index (LAI). It's mentioned  that higher resolution data on LAI would probably lead to better predictions of ET. However, there is no indication of the relevance of such improvement. Why should one try to make a better model of ET that in itself is a modelled parameter? It's no surprise that a remotely sensed feature related to forest structure (like LAI) can be successfully used to model another model (here the algorithm used to model ET) that in itself contains remotely sensed features that are related to vegetation. But why this would be a relevant result? For various applications, a precise estimate of ET is useful but this study doesn't, rightly so, even claim to improve such estimates. I simply can't see a situation, theoretical or practical, where one would need to model the modelled ET. Perhaps such a situation does exist and if so, the authors would make a big favor for the reader by clearly indicating its existence.